# Smartphone App with an Accelerometer Enhances Patients’ Physical Activity Following Elective Orthopedic Surgery: A Pilot Study

**DOI:** 10.3390/s20154317

**Published:** 2020-08-02

**Authors:** Hanneke C. van Dijk-Huisman, Anouk T.R. Weemaes, Tim A.E.J. Boymans, Antoine F. Lenssen, Rob A. de Bie

**Affiliations:** 1Department of Physical Therapy, Maastricht University Medical Center, 6229 HX Maastricht, The Netherlands; anouk.weemaes@mumc.nl (A.T.R.W.); af.lenssen@mumc.nl (A.F.L.); 2CAPHRI School for Public Health and Primary Care, Maastricht University, 6200 MD Maastricht, The Netherlands; ra.debie@maastrichtuniversity.nl; 3Department of Orthopaedic Surgery and Traumatology, Maastricht University Medical Center, 6229 HX, Maastricht, The Netherlands; t.boymans@mumc.nl; 4Department of Epidemiology, Faculty of Health, Medicine and Life Sciences, School for Public Health and Primary Care, Maastricht University, 6211 LK Maastricht, The Netherlands

**Keywords:** activity monitoring, physical activity, functional recovery, hospitalization, mHealth, wearable sensors, arthroplasty, physiotherapy

## Abstract

Low physical activity (PA) levels are common in hospitalized patients. Digital health tools could be valuable in preventing the negative effects of inactivity. We therefore developed Hospital Fit; which is a smartphone application with an accelerometer, designed for hospitalized patients. It enables objective activity monitoring and provides patients with insights into their recovery progress and offers a tailored exercise program. The aim of this study was to investigate the potential of Hospital Fit to enhance PA levels and functional recovery following orthopedic surgery. PA was measured with an accelerometer postoperatively until discharge. The control group received standard physiotherapy, while the intervention group used Hospital Fit in addition to physiotherapy. The time spent active and functional recovery (modified Iowa Level of Assistance Scale) on postoperative day one (POD1) were measured. Ninety-seven patients undergoing total knee or hip arthroplasty were recruited. Hospital Fit use, corrected for age, resulted in patients standing and walking on POD1 for an average increase of 28.43 min (95% confidence interval (CI): 5.55–51.32). The odds of achieving functional recovery on POD1, corrected for the American Society of Anesthesiologists classification, were 3.08 times higher (95% CI: 1.14–8.31) with Hospital Fit use. A smartphone app combined with an accelerometer demonstrates the potential to enhance patients’ PA levels and functional recovery during hospitalization.

## 1. Introduction

Elective joint replacement is an effective and successful intervention in patients suffering from end-stage osteoarthritis [1,2,3,4]. Due to the rising prevalence of osteoarthritis, the number of total knee arthroplasty (TKA) and total hip arthroplasty (THA) procedures performed annually is increasing steadily. The American Joint Replacement Registry (AJJR) shows that 139,582 primary TKA procedures and 93,122 primary THA procedures were undertaken in the United States of America in 2018 [5].

The perioperative care process around TKA and THA procedures has greatly improved in recent years due to advances in surgical techniques and the introduction of clinical care pathways [6,7,8]. Clinical care pathways are directed at preparing the patient for discharge as soon as possible after surgery, without compromising outcomes [3,8,9]. Postoperative mobilization on the day of surgery within a pathway-controlled fast-track program is associated with a reduced length of stay (LOS), enhanced functional recovery, reduced pain, and lower mortality rates [8,10,11].

Although the beneficial effects of physical activity (PA) during hospitalization are well documented, patients continue to spend between 92% and 96% of their time lying or sitting [12,13,14]. Therefore, strategies aimed at increasing the amount of time spent standing and walking are needed [15]. Postoperative physiotherapy is aimed at enhancing PA levels and functional recovery of activities of daily living which are essential in order to function independently at home [9,16,17,18]. However, physiotherapists lack objective insight into the amount of time patients are active. In order to advise patients effectively on their PA behavior, continuous PA monitoring with real-time feedback should be implemented in standard care.

mHealth could provide a solution to this issue [19]. mHealth has been defined by the WHO as “medical and public health practice supported by mobile devices, such as smartphones, tablets or wireless patient-monitoring sensors” [19,20]. mHealth has been developed for many different purposes. It has been applied for the management of blood pressure, management of glucose levels, fall detection, mental health, medication management and PA monitoring [19,21]. PA can be monitored by connecting external wearable devices such as accelerometers, gyroscopes or pedometers, to a smartphone or tablet via Bluetooth [21,22]. Wearable sensors can also be embedded in a smartphone or smartwatch [21,23]. These remote measurement technologies enable continuous PA monitoring and have the advantage of providing patients and healthcare providers real-time feedback. Prior studies have demonstrated that smartphone applications combined with an activity tracker are able to increase the amount of PA of the user [24,25]. Depending on the intended use, additional functionalities such as educational material, exercise programs or capturing patient reported outcomes (PROMs), could also be added to mHealth tools to enhance the possibilities.

mHealth tools support the prevention and treatment of low levels of PA as well as stimulate functional recovery. They have the potential to increase patient awareness, support personalized care, and stimulate self-management. Furthermore, they can motivate patients in the absence of healthcare providers and make them more active and effective managers of their recovery [24,26,27].

Within the orthopedic rehabilitation pathway, mHealth tools are being used to monitor PA in support of outpatient physiotherapy [28,29,30]. The use of mHealth to monitor PA has also been shown to be beneficial to other areas of research. mHealth tools have demonstrated their ability to stimulate the PA of patients with coronary heart disease (CHD) [23,31], chronic obstructive pulmonary disease (COPD) [26], type II diabetes [26] and can motivate the elderly to undertake PAs when implemented in a care home [19].

So far, no mHealth tool is available that offers hospitalized patients and their physiotherapists essential strategies to enhance their PA levels and support their recovery process. Most accelerometry-based activity monitors are validated in healthy adults and lack the sensitivity to measure slow gait [32,33]. Due to the frequent use of walking aids as well as slow and impaired gait, the algorithm of most of the available activity monitors is not validated in terms of being used in hospitalized patients. Therefore, the Department of Physiotherapy of Maastricht University Medical Center (MUMC+) and Maastricht Instruments B.V. developed Hospital Fit (HFITAPP0, Maastricht Instruments B.V., the Netherlands). Hospital Fit is designed to be used in hospitalized patients and consists of a smartphone application connected to an accelerometer. The algorithm of the accelerometer has been validated to differentiate lying and sitting from standing and walking in hospitalized patients [34,35,36]. It provides patients and physiotherapists feedback on the number of minutes spent standing and walking per day. Additionally, it provides patients insight into their own recovery progress, and a tailored exercise program supported by videos. Hospital Fit has been implemented in the standard physiotherapy treatment of patients following TKA and THA in Maastricht University Medical Center since February 2019.

The primary aim of this pilot study was to get a first impression of whether introducing Hospital Fit as part of standard physiotherapy treatments has led to a change in the amount of PA of hospitalized patients who had undergone elective TKA or THA. The secondary aim was to explore whether Hospital Fit has led to a change in the time until functional recovery is achieved in this population.

## 2. Materials and Methods

### 2.1. Study Design

This single center pilot study, with a non-randomized quasi-experimental design, was conducted at Maastricht University Medical Center in Maastricht, the Netherlands, between January 2017 and May 2019.

### 2.2. Study Population

Patients scheduled for an elective TKA or THA at the orthopedic ward of Maastricht University Medical Center were invited to participate. Patients scheduled for surgery between January 2017 and December 2018 were recruited for the control group. During this period, Hospital Fit was being developed. Due to the limited number of accelerometers available, only one patient per week was recruited. In December 2018, the development of Hospital Fit was completed. A one-month implementation phase followed in January 2019, during which no patients were enrolled. Patients scheduled for surgery between February 2019 and May 2019 were recruited for the intervention group. After the implementation phase, sufficient accelerometers had become available, enabling the recruitment of consecutive patients in the intervention group. No other changes were made in the clinical care pathway during the study period.

Patients received verbal and written information about the study at preoperative physiotherapy screenings, scheduled six weeks before surgery. A research physiotherapist contacted the patients again on the day of their surgery, and written informed consent was obtained before study initiation. The confidential processing of data and anonymity were guaranteed.

Patients were eligible if they met the following inclusion criteria: receiving physiotherapy after elective TKA or THA, able to walk independently two weeks prior to surgery as scored on the functional ambulation categories (FAC > 3) [37], they were expected to be discharged to their own home, aged 18 years and older, and had a sufficient understanding of the Dutch language. Exclusion criteria were: the presence of contraindications to walking or wearing an accelerometer on the upper leg, admission to the intensive care unit, impaired cognition (delirium/dementia) as reported by the attending doctor, a life expectancy of less than three months, and previous participation in this study. This study was performed in compliance with the Declaration of Helsinki and was approved by the Medical Ethics Committee of the University Hospital Maastricht and Maastricht University (METC azM/UM), registration number 2017-0175.

### 2.3. Procedure

Patients were enrolled after signing informed consent. All patients followed a standardized clinical care pathway for TKA or THA. Preoperatively, paracetamol, gabapentin, naproxen and a gastric protector were administered. Surgery was performed under spinal or general anesthesia in combination with a local infiltration analgesia (ropivacaine, morphine-sulphate, adrenaline). In TKA procedures, a medial parapatellar approach was used with a posterior stabilized implant. In THA procedures, a posterior approach of the hip joint was used. Pain medication was continued until discharge—with the addition of oxycodone. Postoperative physiotherapy was administered to all participating patients, starting within four hours after surgery. The physiotherapy treatment was aimed at increasing PA levels and enhancing functional recovery. Patients received physiotherapy twice daily (30 min per session) until functional recovery was achieved, as measured with the modified Iowa Level of Assistance Scale (mILAS) [38]. Patients in the control group received postoperative physiotherapy and had their PA levels monitored with an accelerometer without receiving feedback. Patients in the intervention group received the same physiotherapy treatment, but Hospital Fit was used in addition.

#### 2.3.1. Device Description

The PA levels were assessed with the MOX activity monitor (MOX; Maastricht Instruments B.V., the Netherlands (Figure 1)). The device contained a tri-axial accelerometer sensor (ADXL362; Analog Devices, Norwood, MA, USA) in a small waterproof housing (35 × 35 × 10 mm, 11 g). Raw acceleration data (±8 g) were measured in three orthogonal sensor axes (X, Y and Z) at a 25 Hz sampling rate. The accelerometer was factory calibrated against gravity for each axis. The raw acceleration data were converted to a PA classification using a previously described embedded algorithm [34]. After sensor noise reduction, the data were segmented in to one-second long windows using a fixed non-overlapping sliding window. Based on the amount of activity, each window was classified as dynamic or static. For the static windows, the sensor orientation was assessed. Based on a cut-off value of 0.8 g the static windows were classified as standing or sedentary. Each minute the classified results were sent to the Hospital Fit smartphone application via a Bluetooth protocol.

#### 2.3.2. Hospital Fit

Hospital Fit consists of a smartphone-based app which is connected to the MOX Activity Monitor via Bluetooth. It contains a separate interface for patients and physiotherapists, enabling extensive options for physiotherapists. During the first treatment, the physiotherapist applied the accelerometer and installed the app on the patients’ smartphone. The physiotherapist subsequently initiated a connection between the accelerometer and the app by starting a new measurement in the physiotherapist interface.

The PA overview provides patients and their physiotherapists real-time feedback on the number of minutes spent standing and walking per day. An overview was provided per day (Figure 2A), with the possibility to look back at the PA levels of previous days. Additionally, a weekly overview was provided to enable the monitoring of the progress in PA levels over time (Figure 2B).

The recovery assessment gave patients the option of gaining insight into their own recovery progress. The extent of functional recovery can be evaluated by the physiotherapist during every treatment. The ability of patients to perform the activities of daily living was scored on the physiotherapist interface based on the mILAS (Figure 3A). The mILAS assesses the ability of patients to perform several activities of daily living (transfer from the supine position to sitting and vice versa, sit-to-stand, walking, and stair climbing) and rates the amount of assistance and type of walking aid needed. The degree of assistance needed to perform each task safely was scored (0–6 points score per item). The total scores range from 0 to 30, with zero reflecting independence for all items. Stair climbing was only assessed if the patient needed to climb stairs at home; otherwise this item was scored as zero [38]. Because accelerometers are not able to measure the amount of assistance or the type of walking aid needed during PA, scoring the extent of functional recovery had to be performed by the physiotherapist. If necessary, the extent of functional recovery could be adapted multiple times per day.

The mILAS-score was transformed into a percentage score in the app, with 100% indicating complete independency. The percentage scores were provided per activity, showing which activities need improvement in order to reach functional recovery. The percentage scores are supported by a graph, showing progress in functional recovery over time (Figure 3B). More detailed information on the amount of assistance needed is provided per activity, supported by a graph showing the progress over time per activity (Figure 3C). The physiotherapist interface enables the extent of functional recovery to be scored as well as providing an overview. The patient interface only provides an overview of the extent of functional recovery.

Furthermore, the physiotherapist interface contains the option of creating a patient-specific exercise program supported by videos. Hospital Fit contains a database of 25 videos aimed at enhancing functional recovery, upper and lower leg strength, and physical fitness (Figure 4). The videos supporting functional recovery were designed especially for hospitalized patients and show patients how to transfer from the supine position to sitting and vice versa, sit-to-stand, walk, and climb stairs with different types of walking aids. The videos supported the physiotherapy treatment and were aimed at stimulating self-management. After each treatment, the physiotherapist selected appropriate videos, thereby creating a personalized exercise program. If preferred, a note containing personalized information on the number of repetitions or intensity of the exercise could be added as well. The physiotherapist could adapt the exercise program as often as necessary. The patient interface enabled the patient to view the videos as often as they preferred. During each treatment session, the physiotherapist and patient evaluated the amount of PA, extent of functional recovery and the exercise program.

### 2.4. Outcome Measures

#### 2.4.1. Physical Activity

The primary outcome measure was the time spent physically active (total number of minutes standing and walking) per day. The time spent standing and walking was considered the most important outcome since hospitalized patients spend large amounts of time lying and sitting. The MOX activity monitor has been validated to differentiate lying and sitting from standing and walking in hospitalized patients. It has a high validity to estimate the time spent on the activities and postures in a controlled laboratory setting and in free-living conditions [35,36].

During the first treatment session, the accelerometer was attached to the upper leg with a hypoallergenic patch (ten centimeters proximal to the patella, on the non-operated leg). The position of the accelerometer is visualized in Figure 5.

PA was monitored 24 h per day. Days with ≥20 h of wear time were considered valid measurement days and were included in the analysis. After the last treatment session, the accelerometer was removed and the raw tri-axial accelerometer data (Figure 6) were uploaded to a computer.

MATLAB (version 9.5 (R2018b) Natick, Massachusetts: The MathWorks Inc.: Natick, MA, USA; 2018) was used to calculate the total number of minutes spent standing and walking per day. A schematic overview of the data processing is shown in Figure 7.

#### 2.4.2. Functional Recovery

The secondary outcome measure was the achievement of functional recovery on postoperative day one (POD1). Functional recovery was assessed by the physiotherapist during each treatment session using the mILAS and was reported in the electronic health record. In the intervention group, it was also reported in the app. The achievement of functional recovery on POD1 was defined as having reached a total mILAS-score of zero on or before POD1, using a dichotomized outcome (0 = mILAS = 0 > POD1; 1 = mILAS = 0 ≤ POD1). The mILAS shows a high reliability, validity and responsiveness when used to measure functional recovery in the acute phase after TKA or THA [38].

The independent variables measured were: Hospital Fit use (control versus the intervention group), age, sex, body mass index (BMI), type of surgery (TKA or THA), and comorbidities assessed by the American Society of Anesthesiologists (ASA) classification (ASA-class ≤ 2 versus ASA-class = 3; a higher score indicates being less fit for surgery). The medical and demographic data measured were the type of walking aid used and LOS, with the day of surgery being defined as day one. All measurements were extracted from the electronic health record.

### 2.5. Sample Size Calculation

Based on a significance level of 0.05, a power of 0.80, an effect size of 0.20, and five determinants, a sample size of *n* = 75 was needed. Due to a lack of representable data available to determine the effect size, it was determined based on Cohen’s rule of thumb, indicating a medium to large effect size [39]. Accounting for a 20% drop-out rate, we aimed to enroll *n* = 94 patients in this study. The ratio between patients included in the control and intervention group was set at 2:1, respectively. The data analysis was performed according to the intention-to-treat principle. Missing values were not substituted and drop-outs were not replaced.

### 2.6. Data Analysis

Descriptive statistics are presented as means, and standard deviations (SD) or 95% confidence intervals (CI) for continuous variables. The median and interquartile ranges (IQR) were used to present not normally distributed data. The frequencies and percentages were used to present categorical variables. A multiple linear regression analysis was performed to determine the association between the time spent physically active per day and Hospital Fit use, corrected for potential confounding factors (age, sex, BMI, ASA-class, and type of surgery). A univariate regression analysis was performed to determine the association between the time spent physically active per day and Hospital Fit use. Next, potential confounding variables were added (enter method) to explore the association between Hospital Fit use and the time spent physically active per day, corrected for confounding variables. Variables that resulted in a ≥10% change in the regression coefficient of the main determinant (Hospital Fit use) were eligible for inclusion in the model. The variable contributing the most was included in the multiple regression model first, followed by the next variable leading to the highest percentage (≥10%) of change in the main regression coefficient. This process was repeated until there were no more potential confounding factors, resulting in the final model [40]. A multiple logistic regression analysis was performed additionally, to determine the association between the achievement of functional recovery on POD1 and Hospital Fit use, corrected for potential confounding factors. The same procedure was performed as in the linear regression analysis and the same potential confounding variables were explored. Assumptions were checked for both regression analyses by residual plots and statistics. For all statistical analyses, the level of significance was set at *p* < 0.05. All statistical analyses were performed using SPSS (version 23.0.0.2; IBM Corporation Armonk, NY, USA).

## 3. Results

In total, 97 patients were willing and able to participate. The baseline characteristics of both groups are listed in Table 1. Of these patients, nine (9.3%) were excluded because of missing data (no valid measurement day of ≥20 h of wear time due to a delayed postoperative fixation of the accelerometer (*n* = 5) or discharge on POD1 (*n* = 1)), and accelerometer malfunctioning (*n* = 3).

This left 88 cases (90.7%) for analysis, 61 (69%) in the control group and 27 (31%) in the intervention group. In the control group, the median age (interquartile range (IQR)) was 67.19 (11.35) years and 46 patients (72%) had undergone TKA. The control group consisted of 23 women (38%) and 38 men (62%). Fifty patients had an ASA-class of 1 to 2 (82%) and 11 patients (18%) had an ASA-class of 3. The median (IQR) LOS was 3.00 (1) days. In the intervention group, the median age (IQR) was 63.73 (16.62) years and 14 patients (52%) had undergone TKA. The intervention group consisted of 16 women (59%) and 11 men (41%). Nineteen patients had an ASA-class of 1 to 2 (70%) and eight patients (30%) had an ASA-class of 3. The median (IQR) LOS was 3.00 (0) days. The missing values were negligible; data on the achievement of functional recovery on POD1 (*n* = 4) were missing. Differences in the baseline characteristics were accounted for in the regression analyses.

A median (IQR) number of 1.00 (0) valid measurement days (≥20 h wear time) was collected. PA data for 84 patients (95%) was available on POD1 (*n* = 61 control group, *n* = 23 intervention group). On postoperative day two (POD2), the majority of patients were discharged (*n* = 61, 69%), and data for only 23 patients (26%) were available (*n* = 17 control group, *n* = 6 intervention group). From postoperative day three until day seven, data of the valid measurement days were available for just one patient (intervention group). Due to the large reduction in valid measurement days from POD2 onwards, data of these days were not included in the analysis.

The results of the univariate linear regression analysis are shown in Table 2. The results show that Hospital Fit use led to an increase of 32.10 (95% CI: 9.35–54.84) minutes standing and walking on POD1. Patients who did not use Hospital Fit stood and walked on average 70.89 (95% CI: 59.00–82.80) minutes on POD1 compared to 102.99 (95% CI: 82.77–123.21) minutes in patients who used Hospital Fit.

To correct for the influence of potential confounders (age, sex, BMI, ASA-class, and type of surgery), the association between Hospital Fit use and the time spent physically active per day was explored. The addition of age resulted in a 11.41% change in the regression coefficient of the main determinant (Hospital Fit use) and was therefore added to the model. The remaining variables were then added to the model corrected for age, but each resulted in a <10% change in the regression coefficient of the main determinant and were therefore not included. The results of the multiple linear regression analysis are shown in Table 3. The results show that, corrected for age, patients who used Hospital Fit stood and walked on average 28.43 min (95% CI: 5.55–51.32) more on POD1 than patients who did not use Hospital Fit. The model shows that an increase in age led to a decrease in the number of minutes standing and walking on POD1.

The results of the univariate logistic regression analysis (Table 4) show that the odds of achieving functional recovery on POD1 were 2.72 times higher (95% CI: 1.05–7.049) for patients who used Hospital Fit than for patients who did not use Hospital Fit.

The influence of potential confounders on the association between the Hospital Fit use and time spent physically active per day was explored. Addition of ASA-class resulted in the largest change of 12.38% in the regression coefficient of the main determinant (Hospital Fit use) and was added to the model. The remaining variables were then added to the model corrected for ASA-class, but each resulted in a < 10% change in the regression coefficient of the main determinant and were therefore not included. The results of the multiple logistic regression analysis (Table 5) show that, corrected for ASA-class, the odds of achieving functional recovery on POD1 were 3.08 times higher (95% CI: 1.14–8.31) for patients who used Hospital Fit than for patients who did not use Hospital Fit. Including ASA-class in the model shows that a lower ASA-class increased the odds ratio for a functional recovery on POD1.

## 4. Discussion

In this pilot study we aimed to gain a first impression of whether Hospital Fit has the potential to improve the amount of PA and time until functional recovery is achieved in hospitalized patients following orthopedic surgery. The results show an increase in the time spent standing and walking, as well as higher odds of functional recovery on POD1 from the introduction of Hospital Fit. Although the guidelines on the recommended amount of PA during hospitalization do not yet exist, an average improvement of 28 min (39%) standing and walking on POD1 can be considered a clinically relevant contribution to prevent the negative effects of inactivity.

The relatively large confidence intervals indicate a large variation in PA levels during hospitalization. These large differences in the PA levels of hospitalized patients are seen in many other studies as well [13,14,41,42]. PA levels can be influenced by many different factors such as symptoms, the motivation of the patient, awareness of the importance of PA, the availability of healthcare staff to assist patients during walking, or the availability of adequate walking aids [15,43,44]. These factors are expected to result in large differences in PA levels between patients.

Wearable technology is increasingly being used in TKA and THA research, with the assessment of PA, functional parameters, and gait analysis as primary modes of investigation. No standard outcome measure or testing methodology has been established in wearable-based PA monitoring following TKA or THA [45]. Technology, testing protocol and sensor-based outcome variables may vary and may affect the quality and reliability of the data being collected [46,47,48,49].

Limited research has been conducted on monitoring PA during the early recovery phase following TKA or THA [45]. Eight studies have been performed, with sensor-based outcome variables varying considerably between studies [28,50,51,52,53,54,55,56]. Two studies investigated the amount of time spent active (standing and walking) as outcome variables in the monitoring of PA of hospitalized patients following TKA [50,51]. Schotanus et al. [50] showed that patients within an enhanced recovery pathway spent 9% of their waking hours standing and walking on POD1. On POD2, this increased towards 11%, with a planned discharge within three days post-operation. PA was measured with a triaxial accelerometer (GC Dataconcets LLC, Waveland, USA) attached to the non-operated thigh. No details were provided regarding the validity of the algorithm to differentiate lying and sitting from standing and walking in hospitalized patients. Due to the lack of insight into the number of waking hours, the results cannot be compared to our study. In agreement with our study, they concluded that accelerometry is an added value for the objective analysis of PA during the early recovery phase in patients after TKA [50]. Fenten et al. [51] compared the amount of time spent active per day between patients receiving periarticular local anesthetic infiltration (LIA), and patients receiving LIA of the posterior knee capsule in combination with a femoral nerve block (FNB) catheter. PA was monitored with an accelerometer, attached to the non-operated thigh. No details were provided regarding the accelerometer type or the validity of the algorithm to differentiate lying and sitting from standing and walking in hospitalized patients. PA was monitored between 8 am and 8 pm on the day of surgery and on POD1. The mean time spent active (SD) on POD1 was 20.5 (14.9) minutes in the FNB group versus 27.7 (14.1) minutes in the LIA group [51]. Although the postoperative physiotherapy treatment and LOS were comparable, our study shows higher amounts of time spent active on POD1. Patients who did not use Hospital Fit spent 70.89 (95% CI: 58.93–82.86) minutes active compared to 102.99 (95% CI: 82.77–123.21) minutes in patients who used Hospital Fit. These differences might be explained by the fact that, in our study, PA was monitored continuously for 24 h per day and patients scheduled for a prolonged stay in an outpatient rehab clinic were excluded. So far, no studies have investigated the amount of time spent active in hospitalized patients following THA.

One of the main aims of Hospital Fit is to decrease the negative effects of sedentary behavior in hospitalized patients through stimulating the amount of time spent active. As hospitalized patients spend over 92% of their time lying or sitting [12,13,14], the number of minutes spent standing and walking per day is deemed the most appropriate sensor-based outcome variable for Hospital Fit. Additionally, it is a practical outcome variable since it is easily interpreted by patients and physiotherapists.

Three studies investigated intensity (activity counts) as an outcome variable in monitoring the PA of hospitalized patients following TKA [52,53,54] and THA [52,53]. We believe however, that monitoring the time spent active is more meaningful than monitoring intensity levels (activity counts) in the early recovery phase after surgery. First, the intensity of PA as perceived by patients may deviate from the intensity measured by the accelerometer. During the first days after surgery, patients may perceive ambulation at low walking speeds as a high intensity activity, while the accelerometer objectively classifies this as a low intensity activity. Second, in the early recovery phase after surgery, the focus of physiotherapy lies on the recovery of activities which are essential to perform at home, such as walking and stair climbing. The focus does not lie on the intensity of the activities performed.

Three studies investigated step counts as an outcome variable in the monitoring of PA of hospitalized patients following TKA [28,55,56] and THA [55]. Using step counts to quantify PA has advantages since it is specific to ambulation and is easily interpreted by patients and physiotherapists [57]. However, during the early recovery phase after TKA and THA, all patients require a walking aid, and slow and impaired gaits are common [58]. Several studies have shown that these factors decrease the validity of activity trackers to measure step counts [59,60,61,62]. Furthermore, movements of the arms or legs performed in bed or on a chair may result in an overestimation of the number of steps taken. We therefore consider the time spent active a more appropriate outcome variable for Hospital Fit than step counts.

The present study investigated an mHealth tool which uses a multimodal approach, tailored specifically to the needs of hospitalized patients and their physiotherapists. Besides objective activity monitoring, Hospital Fit also provides patients insight into their recovery progress and offers physiotherapists the option of creating a patient-specific exercise program supported by videos. Recently, an increasing number of other mHealth tools have been investigated within the orthopedic rehabilitation pathway [28,29,30,63,64,65,66,67,68,69]. These tools are predominantly prescribed in support of outpatient physiotherapy. Besides monitoring PA, they are being used to offer biofeedback in exercise programs, monitor the range of motion (ROM) of the knee joint, capture PROMs, provide educational material and enable telerehabilitation.

Inertial measurement units (IMUs) contain accelerometers paired with gyroscopes and magnetometers, to provide a detailed analysis of limb movements and orientations within a spatial reference frame [45]. The use of IMUs enables patients to receive feedback on the performance of their exercise technique based on supervised machine learning. It also enables counting exercise repetitions as well as recording the ROM of the knee joint. These options can offer additional motivation and feedback to enhance adherence, and can positively impact the patient experience and clinical outcome. Although this technology seems promising, there is a need for such systems to demonstrate a real-world accuracy validation [70,71].

Furthermore, some mHealth tools describe using the internal proprietary algorithm of the patient’s smartphone to passively measure their step count [28,29]. This requires patients to carry their smartphones with them in order to not make an underestimation of the amount of PA performed. During hospitalization however, patients often wear hospital gowns or pajamas without pockets, or leave their smartphones on their nightstand. Therefore, monitoring PA through a smartphone is not recommended in hospitalized patients. Hospital Fit is equipped with an accelerometer attached to the upper leg, and the algorithm is able to differentiate lying and sitting from standing and walking in patients using walking aids, or with slow or impaired gait. This is an advantage over many smartphones and commercially available activity trackers and one of the reasons Hospital Fit was developed.

This study was not without limitations. We acknowledge that with the current study design, the results may not only be attributable to the introduction of Hospital Fit. The current design enabled us to effectively use the time in which Hospital Fit was developed to include patients in the control group, and give us a first impression of the potential of Hospital Fit. Unfortunately, due to technological challenges, the development of Hospital Fit took longer than anticipated. Although the clinical care pathway and physiotherapy treatment did not change during this period, awareness on the importance of PA during hospitalization might have increased among patients and healthcare professionals, which may have resulted in a bias in favor of the intervention group. This could have led to a slight overestimation of the results.

Additionally, the individual functionalities of Hospital Fit were not investigated in this pilot study. Therefore, we cannot establish the relationship between each functionality and PA. Enabling this would have provided valuable information regarding the contribution of the different functionalities on the influence of Hospital Fit.

Furthermore, when this study was designed in 2016, the median (range) postoperative LOS was 6 (4–10) days. The implementation of a new clinical care pathway in November 2016 has resulted in a reduction in the median (range) LOS to 4 (3–12) days, leaving a relatively short period to introduce and use Hospital Fit [8]. In our study, the majority of patients were discharged on POD2 and data for only 23 patients were available on POD2. Because the data of these remaining 23 patients were not representative of the whole population and resulted in insufficient power to perform a regression analysis, only data on the amount of PA performed on POD1 were included in the analysis.

The present study has a number of important implications for daily practice and future research. First, the results show that Hospital Fit has the potential to enhance the amount of PA and functional recovery in hospitalized patients, especially when the hospital stay permits the use of the application for a longer period. Second, since the literature on the amount of PA performed in hospitalized patients following TKA and THA is scarce, this study contributes to the knowledge of the PA behavior of this population. Third, continuous objective monitoring provides patients and their physiotherapists the advantage of being able to set goals regarding the amount of PA. However, reference values regarding the optimal amount of PA after surgery do not exist yet. Hospital Fit and the data it creates have tremendous potential, because continuous PA monitoring as part of standard care will enable creating population norms for PA.

In order to determine the effectiveness of Hospital Fit, it is recommended that this pilot study should be followed by a larger, cluster randomized controlled trial in a population of hospitalized patients with a longer LOS. In order to determine the effect of each functionality of Hospital Fit on PA, investigating the individual functionalities is recommended as well.

## 5. Conclusions

This pilot study aimed to gain a first impression of whether Hospital Fit has the potential to improve the amount of PA and shorten the time until functional recovery is achieved in hospitalized patients following orthopedic surgery. The results show an increase in patients’ time spent standing and walking, as well as higher odds of functional recovery on POD1 due to the introduction of Hospital Fit. This study shows that a smartphone app combined with an accelerometer demonstrates potential to enhance patients’ PA levels and recovery processes during hospitalization.

## Figures and Tables

**Figure 1 sensors-20-04317-f001:**
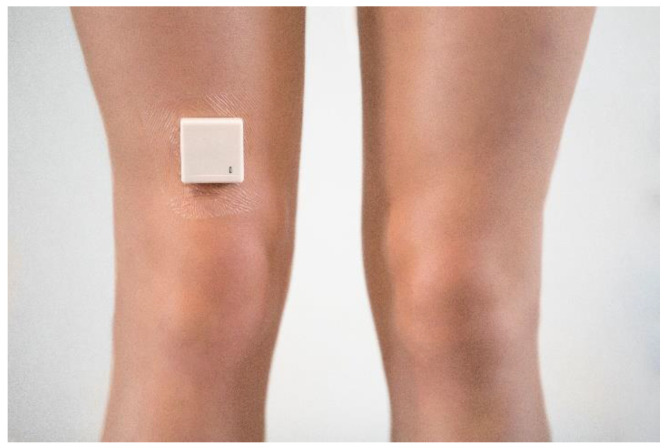
The MOX activity monitor.

**Figure 2 sensors-20-04317-f002:**
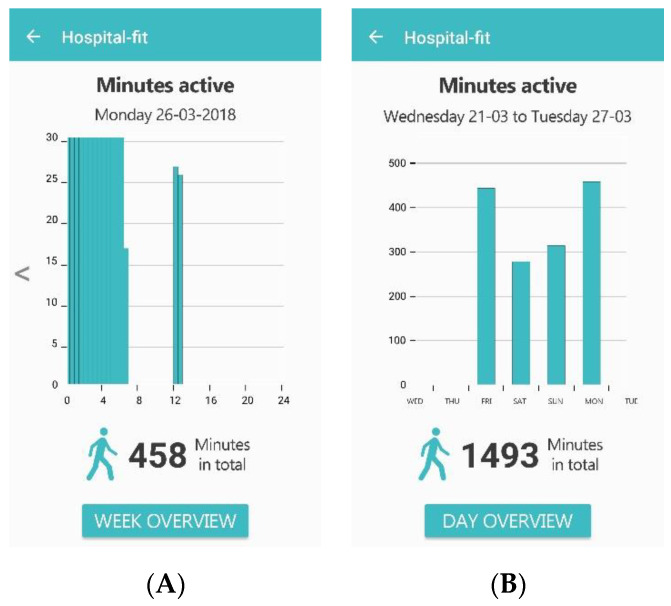
Overview of the total number of minutes spent standing and walking per day (**A**) and per week (**B**).

**Figure 3 sensors-20-04317-f003:**
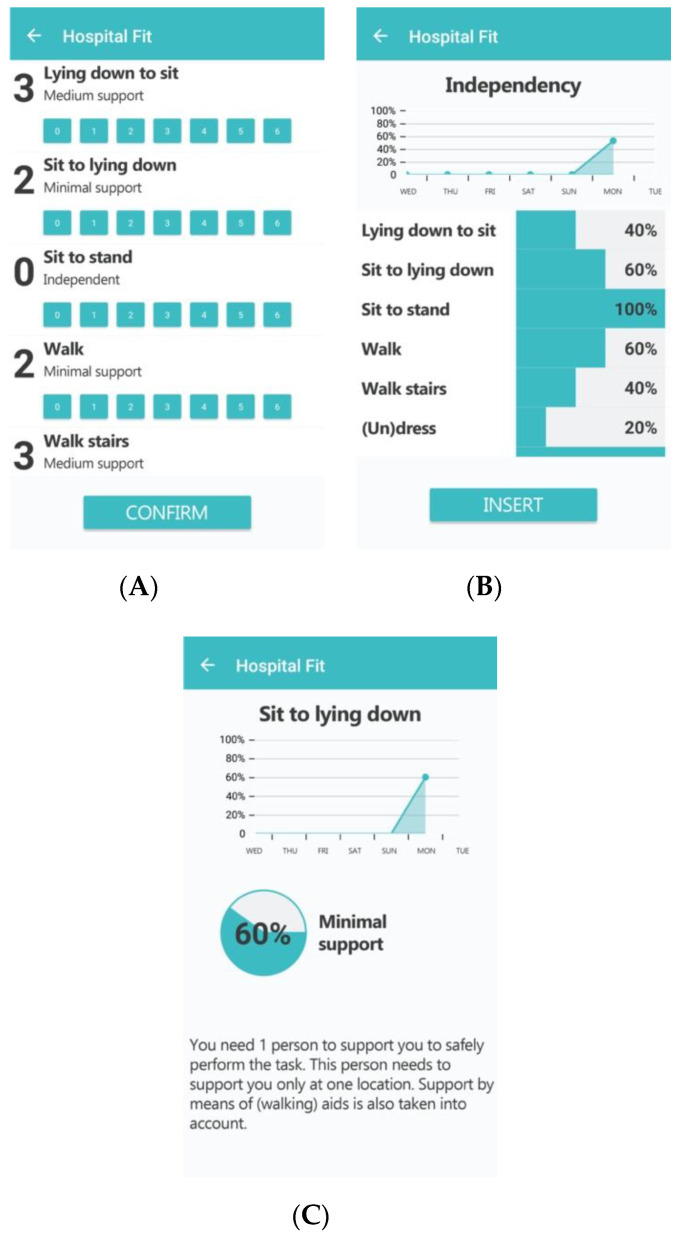
Recovery assessment with the option of scoring the extent of functional recovery based on the modified Iowa Level of Assistance Scale (**A**); an overview of the extent of functional recovery (**B**); the amount of assistance needed and progress over time per activity (**C**).

**Figure 4 sensors-20-04317-f004:**
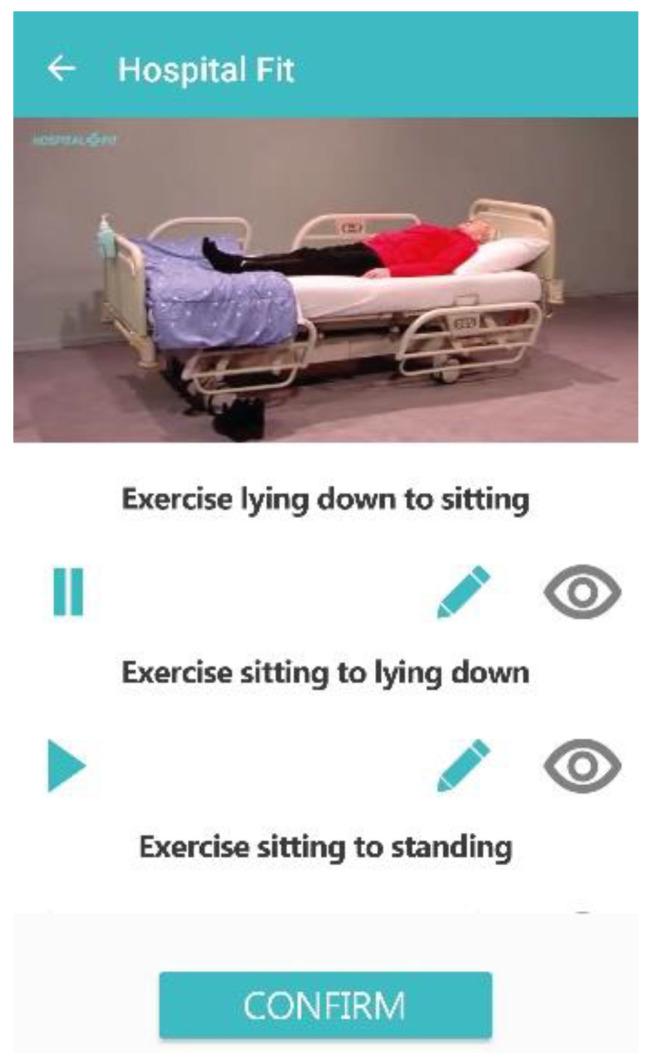
Exercise videos.

**Figure 5 sensors-20-04317-f005:**
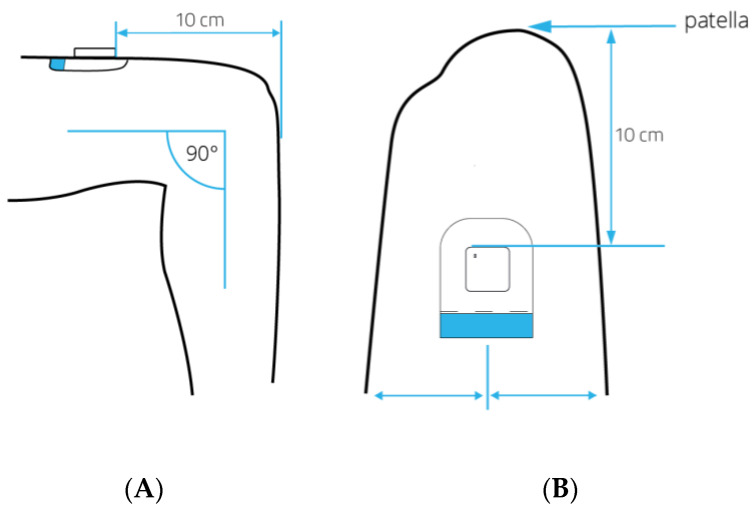
Lateral view (**A**) and frontal view (**B**) of the placement of the MOX activity monitor with the patient in a seated position. Arrows indicate the location of the hypoallergenic patch and sensor on the upper leg, which is 10 cm proximal to the patella.

**Figure 6 sensors-20-04317-f006:**
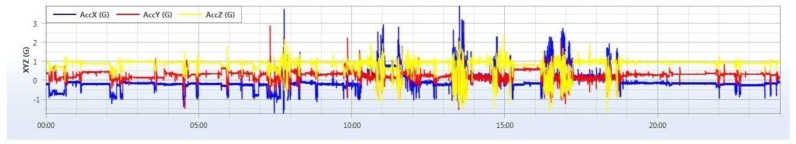
Example of the raw tri-axial accelerometer data of one subject for one measurement day. One measurement day (24 h) is represented on the *x*-axis. G-forces per sensor axes (X, Y and Z) are represented on the *y*-axis.

**Figure 7 sensors-20-04317-f007:**
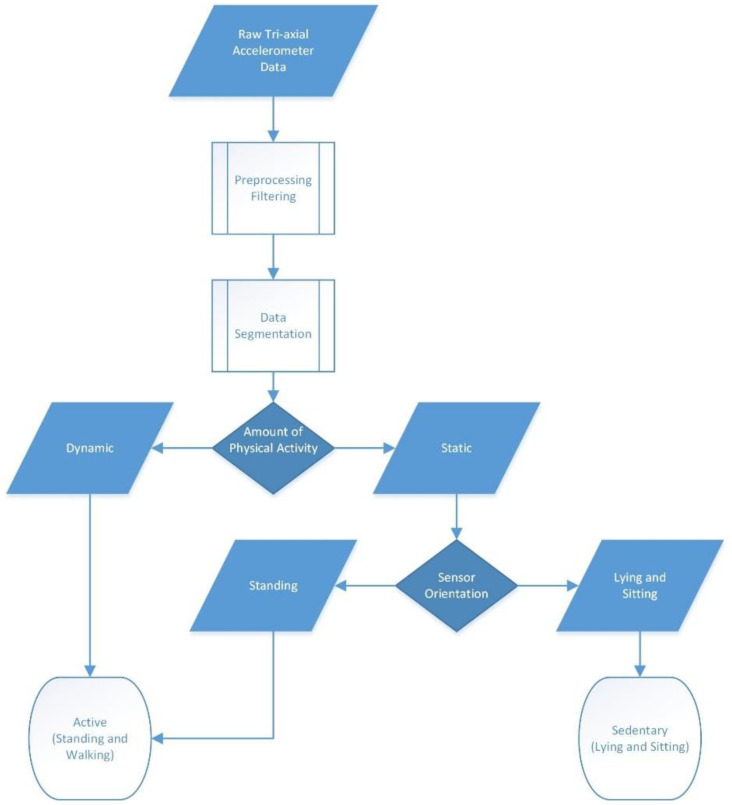
Data processing—a schematic overview of the physical activity classification algorithm for the accelerometer worn on the upper leg location.

**Table 1 sensors-20-04317-t001:** Characteristics of study participants.

	Control Group (*n* = 64)	Intervention Group (*n* = 33)
Age, years (median, IQR)	66.60 (10.62)	65.10 (13.72)
Sex (*n*, %)		
Female	24 (38)	18 (55)
Male	40 (63)	15 (45)
BMI, kg/m^2^ (median, IQR)	27.73 (4.72)	27.47 (4.70)
Type of surgery (*n*, %):		
Total knee arthroplasty	49 (77)	15 (45)
Total hip arthroplasty	15 (23)	18 (55)
ASA-class (*n*, %):		
ASA 1–2	53 (83)	26 (79)
ASA 3	11 (17)	7 (21)
Walking aid (*n*, %):		
Two crutches	53 (83)	31 (94)
One crutch	1 (2)	-
Walking frame	5 (8)	1 (3)
Walker	5 (8)	1 (3)
LOS, days (median, IQR)	3.00 (1)	3.00 (0)

IQR = interquartile range, BMI = body mass index, ASA = American Society of Anesthesiologists, LOS = length of stay in hospital, with the day of surgery being defined as day one.

**Table 2 sensors-20-04317-t002:** Univariate linear regression analysis—the association between the time spent physically active on postoperative day one (POD1) and Hospital Fit use.

	B	Std. Error	*p*-Value	95% Confidence Interval for B	
				Lower Bound	Upper Bound
Constant	70.89	5.98	0.000	59.00	82.80
Hospital Fit use	32.10	11.43	0.006	9.35	54.84

POD1 = postoperative day one.

**Table 3 sensors-20-04317-t003:** Multiple linear regression analysis—the association between the time spent physically active on POD1 and Hospital Fit use, corrected for age.

	B	Std. Error	*p*-Value	95% Confidence Interval for B	
				Lower Bound	Upper Bound
Constant	124.25	31.80	0.000	60.98	187.52
Hospital Fit use	28.43	11.50	0.016	5.55	51.32
Age	−0.81	0.48	0.092	−1.76	0.13

POD1 = postoperative day one.

**Table 4 sensors-20-04317-t004:** Univariate logistic regression analysis—the association between the achievement of functional recovery on POD1 and Hospital Fit use.

	B	Std. Error	*p*-Value	Odds Ratio	95% Confidence Interval for Odds Ratio	
					Lower Bound	Upper Bound
Constant	−0.31	0.26	0.243	0.735	-	-
Hospital Fit use	1.00	0.49	0.039	2.720	1.050	7.049

POD1 = postoperative day one.

**Table 5 sensors-20-04317-t005:** Multiple logistic regression analysis—the association between the achievement of functional recovery on POD1 and Hospital Fit use, corrected for ASA-class.

	B	Std. Error	*p*-Value	Odds Ratio	95% Confidence Interval for Odds Ratio	
					Lower Bound	Upper Bound
Constant	−0.91	0.58	0.112	0.401	-	-
Hospital Fit use	1.13	0.51	0.026	3.080	1.14	8.31
ASA-class	0.71	0.59	0.228	2.03	0.64	6.39

POD1 = postoperative day one, ASA = American Society of Anesthesiologists.

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
