# Peer review of "Smartphone App with an Accelerometer Enhances Patients’ Physical Activity Following Elective Orthopedic Surgery: A Pilot Study"

_sensors, 2020, doi:10.3390/s20154317_

Round 1
Reviewer 1 Report
The authors present an interesting pilot Randomised Controlled Trial exploring the use of a smartphone application (called Hospital Fit) to monitor patients’ physical activity after elective surgery. The article is of great interest; however, the authors should provide more detailed justifications for their choices and provide more information as to the background and context. Hereafter is a detailed list of comments (generally in order of appearance in the manuscript).
- Line 40: As the authors have submitted to a Journal which has a large international readership, it would be useful to provide an international comparison to the types of procedures performed and potentially the number of cases.
- Introduction: In general, the introduction provides a fairly concise overview of THA/TKA, however little attention is focused on the utility of technology, and how it has been used in other areas of research (such as mental health, other physical health areas). I would recommend providing a brief summary of some key items of work, and possibly reference the term Remote Measurement Technology.
- Study approach: While this is a pilot RCT the authors should provide more details surrounding the approach, for example, were the study team blinded to randomisation? Were participants? How did randomisation take place? Did the authors make any bias into account when performing analysis? Was the pilot registered with any service?
- Why were participants administered drugs such as Paracetamol, Gabapentin, Naproxen and a Gastric protector? It is not clear if this is being provided for context, or as it influenced study outcomes.
- Line 161-162: For clarify, is the total number of steps determined even if a participant only made one step in a minute? If so, how did the authors account for noise?
- For functionality recovery, and the use of MILAS, has this been validated? Why was it chosen over other measures?
- Sample size: More detailed is needed to understand the power calculation. Why was a 0.20 determined, and is this effect for change in activity levels?
- Table 1: Did the authors perform any sensitivity analysis to explore differences between the groups?
- Table 2: The CI are very large between the samples and the authors spend little time discussing this variation. It would be helpful for the authors to explore it further.
- Discrepancy: The authors indicate in the discussion a limitation surrounding the study being designed in 2016, yet recruitment occurred in 2017, is the point being: A change in national policy means patient contact time is reduced which makes data collection more difficult?
My comments are intended to support the authors and should be in no way seen to detract from the high quality work.
Reviewer 2 Report
This paper deals with a developed smartphone app that monitors the physical activities of patients post TKA or THA surgery. The authors have prepared the paper without many grammatical or format errors. However, I do have concerns concerning the originality and main objective of this work. It is hard to understand how this app compares to other physical activity apps, and the explanation of the basic function of the Hospital Fit are missing.
Key methodological explanations are missing and the results of the study are only visible in just three tables, one of which is a list of study participants.
Line 121
Explanation of the Hospital Fit is too vague, suggest revising this section. Since the novelty of this app is one of the scopes of this paper, the basic function on how this app operated should be explained. Though Fig. 1 is provided as reference, I feel that this just thumbnail of the app with little significance to the reader. Please provide additional explanation, including figures, to how this app was specifically used in this study.
For example, how could the patients gain insight into their own recovery progress? What kind of videos were shown to support the exercise program? Include a flow chart of the exercise program protocol?
Line 148
These is no mention to what kind of data was measured. How was the accelerometer data converted into meaningful PA data? I know that this is not the main scope of this study, however the reader would like to know the basic methodology in how the data was processed.
These should include:
- Sensor specifications (measurement range, measurement time, sampling rate, data transfer protocols, etc)
- Procedure explaining how raw (tri-axial?) accelerometer data was processed. Was the magnitude of the tri-axial acceleration data used or individual axes separately processed?
- How was the threshold set to detect specific PA inputs? Was the reliability of this input validated?
- Fig.2 lacks information. What does the right part of the figure indicate?
- What kind of calibration procedures were implemented after attaching the sensors?
Line 218
The results are summated into just two tables. Additional follow up result data should be included to explain how the authors were able to reach the data to these tables. For example including additional data to indicate how each of the PAs were segmented from the whole data.
Round 2
Reviewer 1 Report
The authors have addressed my concerns.
Author Response
"Please see the attachment"

Reviewer 2 Report
The authors have made the effort to revise the manuscript, but I do have some concerns of the content.
General Comments:
The authors have put a lot of effort in revising this paper. However there are some concerns regarding the content to which this manuscript is prepared.
- Figures should be revised.
- Which function of the Hospital Fit contributed to the overall increase in the PA?
- The revised manuscript shows the correction made but it is hard to distinguish what is the final edit.
Throughout the manuscript
Information concerning the experimental setup, protocols and how the data was processed have been greatly increased compared to the previous version.
Reading from the improved manuscript, I can now summate that Hospital Fit has the following functions:
- Real time feedback of standing and walking per day
- History check of previous PA
- Personal recovery check
- PA scoring
- Exercise support via video
- Personalized exercise program
Versus the control group received only:
- Postoperative physiotherapy
In the conclusion the authors state that :
“The results show an increase in time spent standing and walking, as well as higher odds of functional recovery on POD1 since the introduction of Hospital Fit. This study shows that a smartphone app combined with accelerometer demonstrates potential to enhance patients’ PA levels and recovery process during hospitalisation”
After reading the results and discussion, I am puzzled to understand what caused the Hospital Fit patients to increase their PA. There are so many functions within the Hospital Fit app that are not individually investigated. For example, the exercise support via video is not necessary a function that requires IMU data. Nor is it a novel function to Hospital Fit. If another group with postoperative physiotherapy + video support might yield better results. Why were more groups with intermediate functions (between control and Hospital Fit) not included in this study? I understand that Hospital Fit is an integration of all these multiple functions, however the critical factor or the most important function that this app has to provide is not described. To investigate this I believe further studies have to be conducted in order to establish the relationship between each function and their affect on PA improvements. Otherwise this app will not be able to distinguish itself from other similar apps in the future with even more functions.
Specific comments below:
Same figure used for Fig 1 and 5. Suggest combining to one.
Is Fig 2 really necessary? This just seems like a thumbnail picture. Suggest removing.
Figure 3 Why are two of the same screenshot overlapping each other? Is this figure even necessary?
Figure 5 requires more caption/information. Where is the patella?
Line 262
“MATLAB (version 9.5 (R2018b) Natick, Massachusetts: The MathWorks Inc.; 2018) was used to calculate the total number of minutes spent standing and walking per day.”
It will be interesting to see what data was used as input and the resultant data. Suggest adding a sample data as reference and a flow chart to simply explain how the data was processed.
Author Response
"Please see the attachment".
